# Nationwide Survey on Seasonal Influenza Vaccination among Health Care Workers during the COVID-19 Pandemic in Greece: Determinants, Barriers and Peculiarities

**DOI:** 10.3390/ijerph20136247

**Published:** 2023-06-28

**Authors:** Ioanna Avakian, Katerina Dadouli, Lemonia Anagnostopoulos, Konstantinos Fotiadis, Athanasios Lianos, Paraskevi Mina, Christos Hadjichristodoulou, Varvara A. Mouchtouri

**Affiliations:** 1Laboratory of Hygiene and Epidemiology, Faculty of Medicine, University of Thessaly, 22 12 Papakyriazi Street, 41222 Larissa, Greece; joavakian@uth.gr (I.A.); adadouli@uth.gr (K.D.); lanagnost@uth.gr (L.A.); atlianos@uth.gr (A.L.); xhatzi@uth.gr (C.H.); 2Hmathia General Hospital, Veria Hospital Unit, 59132 Veria, Greece; kostasfotiad@yahoo.gr

**Keywords:** seasonal influenza vaccination, seasonal influenza, vaccination, health care workers, COVID-19 vaccination, vaccines

## Abstract

Background: Seasonal influenza vaccination (SIV) of health care workers (HCWs) is critical in protecting patients’ and HCWs’ health. Our objective was to examine HCW SIV coverage and related determinants. Methods and Materials: A nationwide cross-sectional questionnaire survey was conducted among HCWs during the first half of 2021. The questionnaire (online or paper-based) included knowledge, attitude and practice questions regarding SIV, COVID-19 vaccines and vaccination. Results: Out of 6500 questionnaires administered, 2592 were completed (response rate: 39.9%). SIV coverage reached 69.4% (95% CI: 67.6–71.2%) based on self-reported vaccine uptake. Nurses and administrative staff were found to be more skeptical and have lower vaccine acceptance in comparison with physicians (aOR = 0.66 and aQR = 0.59, respectively). Other SIV hesitancy risk factors included working in secondary health care (aOR = 0.59) and working in northern Greece (aQR = 0.66). Determinants for SIV acceptance included being or living with high-risk people due to medical history (aOR = 1.84 and aOR = 1.46, respectively), positive attitudes towards routine vaccinations (aOR: 1.86), knowledge about COVID-19 vaccines (aOR = 1.53) and COVID-19 vaccine uptake (aOR = 3.45). The primary reason for SIV refusal was low risk perception (58.7%). Conclusions: SIV coverage (2020/2021) was relatively high, but remained far from formal recommendations. Specific occupational groups were skeptical and low-risk perception was the main reason for vaccine refusal. Targeted policies should be developed and enforced.

## 1. Introduction

Seasonal influenza vaccination (SIV) has proven to be a significant public health measure, saving numerous lives and effectively preventing workplace absenteeism [1,2,3,4]. Before the emergence of the coronavirus pandemic, global mortality from the influenza virus was estimated at 250,000–500,000 deaths per year, while other sources reported even greater levels of up to 700,000 deaths annually [5]. In the European Union/European Economic Area (EU/EEA), seasonal influenza may cause up to 50 million symptomatic cases annually, and 15,000–70,000 Europeans die from influenza-related causes [6]. In 2017, the global economic burden from influenza was calculated to be approximately USD 87.1 billion [1].

The importance of SIV for health care workers (HCW) as a preventive measure is twofold, providing benefits to both patients and health professionals [1,7,8,9]. A significant proportion of patients seeking health care are prone to severe illness from influenza due to comorbidities. Secondary bacterial pneumonia is a common complication associated with age and immunocompromised status, along with pre-existing medical conditions [10,11,12,13]. HCWs comprise an occupational group with increased exposure to the virus, and therefore have an increased chance of illness and transmission [14,15]. Moreover, workplace absenteeism of HCWs due to illness during seasonal influenza epidemic periods contributes to the health system’s financial burden and causes staff shortages and reduced efficiency [3]. Ecological studies conducted in the United Kingdom indicate a 10% decrease in absenteeism during influenza season for each 10% increase in HCWs’ immunization rates [2].

SIV rates of HCWs vary largely worldwide. The vaccination of HCWs in most EU member states remains significantly below the World Health Organization’s (WHO) proposed target of 75% coverage [1]. According to the most recent (2018) report from the European Centre for Disease Prevention and Control (ECDC), SIV coverage rates ranged from 15.6% in Italy to 63.2% in Belgium, with a median coverage of approximately 30.2% [16]. North American vaccination coverage rates reached 75.9% for health care personnel in the United States and 71.8% in Canada [17,18]. Based on the limited data available for lower-socioeconomic status countries, low vaccination coverage is indicated in most cases [19]. The suboptimal vaccination rates seen globally are a major public health issue requiring attention and consideration of factors that limit SIV acceptance by the population [1].

Studies regarding vaccination hesitancy identify a variety of factors contributing to public distrust in vaccines. The primary reasons for public skepticism include concerns related to side effects and overall safety, the influence of conspiracy theories combined with a lack of trust in public institutions and misguided beliefs about vaccine necessity, as well as cultural factors, mainly in central Asia. These reasons also extend to HCWs [20]. The COVID-19 pandemic highlighted the objections of a specific population group to vaccination in general; in many studies, COVID-19 vaccine hesitancy was positively associated with reluctance towards other vaccines, including SIV [21,22]. Several factors have also been associated with HCWs’ refusal of SIV, among them occupation and low risk perception [19].

As demonstrated by previous studies, the percentage of vaccinated HCWs in Greece is far from what is required, despite a steady increase over the last decade [1,23,24,25]. Similarly, COVID-19 vaccination hesitancy among HCWs has proven quite resistant. A notable proportion of health care professionals remained unvaccinated despite the mandatory regulation, preferring to be suspended from work for several months rather than receive the COVID-19 vaccine [26]. According to National Registry, the highest proportion of SIV among health care professionals (44.6% for hospital professionals and 67.6% for primary health care professionals) was recorded in the period 2020–2021 [24]. It is likely that lifestyle changes and informational initiatives emerging from the COVID-19 pandemic contributed to creating a more receptive environment regarding SIV [27]. An interesting finding from studies conducted in Greece reflected a positive correlation between HCW COVID-19 vaccination intention and that of the general population. Significant differences were seen for both groups by health district, indicating that HCWs could serve as role models for the general population [28,29].

Our study intended to examine SIV coverage and related determinants of HCWs providing services in both the primary and secondary health care sectors in Greece. Primary health care (PHC) is largely provided by rural health centers (RHC) and urban health units (UHU), which are both public institutions, while secondary health care (SHC) is provided by hospitals (HO), which can be public or private, as well as university-based or general. The objectives of our survey were (a) to determine SIV coverage of HCWs and (b) to examine factors related to the acceptance or refusal of SIV in order to collect useful information that could be used by stakeholders for public health policies.

## 2. Materials and Methods

### 2.1. Study Design

A cross-sectional knowledge, attitudes and practices (KAP) survey was designed. The cross-sectional questionnaire-based KAP survey was conducted from February to June 2021 in PHC and SHC facilities in Greece to assess the knowledge, attitudes and practices of RHC/UHU and HO personnel. During the study period, authorized COVID-19 vaccines were available in Greece. Moreover, questions related to COVID-19 vaccination were included in the questionnaire.

A sample size of 916 RHC/UHU personnel and 1045 HO workers was calculated using the Raosoft Digital Sample Size Calculator [30], with 3% as a margin of error, 95% as the confidence interval (CI) and 50% as the expected frequency, as well as 6456 and 50,000 as the population size for RHC/UHU and HO professionals, respectively [31]. Due to the distribution format and workload of personnel during the study period, the estimated response rate was approximately 30%. A sample of 3000 was calculated for RHC/UHU and 3500 for HO. A geographically stratified sampling plan was applied to produce a representative sample of 125 RHC/UHU and 20 HO, including one general and one university HO, where applicable, in each of the seven health regions. In each RHC/UHU, respondents had the option of completing an online or paper-based questionnaire, while HO personnel completed a paper-based questionnaire.

An anonymous questionnaire consisting of 20 questions was designed in accordance with WHO guidelines, advice issued by the Hellenic National Public Health Organization (NPHO) and relevant studies conducted in Greece [23,32,33]. The questionnaire was developed by an expert team consisting of an epidemiologist, an occupational health professional and a public health specialist. A total of 20 RHC/UHU and HO professionals participated in pilot testing of the questionnaire, helping the research team to estimate the time required for questionnaire completion. Pilot testing further helped determine the clarity of questions addressed to target population and the functionality of the online tool. Pilot questionnaires were used to develop the final form of the survey questionnaire; however, they were excluded from the final analysis.

The time required to complete the questionnaire was approximately 15 min. Procedures for survey data collection, entry, analysis and storage were conducted in accordance with anonymity, privacy and confidentiality regulations of national legislation and rules of the University of Thessaly, Greece.

The survey tool included questions about KAP around vaccination in general and SIV specifically (Appendix A). The first (general) section of the questionnaire included 11 questions regarding demographics; the second (specific) section included nine KAP questions concerning vaccination. Questions about respondents’ and their family’s vulnerability due to medical history and about COVID-19 vaccination intention were included (answer: yes/no). The survey included three questions about knowledge and three about attitudes/perceptions towards routine vaccines, which could be answered on a five-level item scale (“completely agree”, “agree”, “neither agree nor disagree”, “disagree” and “completely disagree”). Respondents were asked to state their practices related to vaccination of their children, if they had any. SIV acceptance was examined by the question “have you been vaccinated with the seasonal influenza vaccine?”. Unvaccinated respondents were asked to explain their reason for refusal through a semi-closed question. One question related to source of information was included: “Which channels do you use to keep informed about the COVID-19 pandemic and the SARS-CoV-2 vaccine, and how often?” (four-level item scale: “always”, “often”, “rarely”, “never”). Respondents were asked about their preferences concerning sources of information related to the COVID-19 pandemic as an indication of a more spontaneous reaction to health-related information searches. Lastly, the specific section of the questionnaire included questions about COVID-19: one regarding knowledge about COVID-19 vaccines and one about respondents’ COVID-19 vaccination status. Unvaccinated participants were asked to specify their reason for refusal (semi-closed question).

### 2.2. Ethical Statement

The survey was approved by the Ministry of Health and the health district authorities. Ethical approval from the scientific committee of the University of Thessaly (protocol number 49 and number 48/13.01.2021) was obtained. All participants provided written consent before completing the questionnaire and were fully informed about the survey’s objectives and personal data safety regulations.

### 2.3. Statistical Analysis

The statistical analysis was carried out using IBM SPSS Statistics for Windows, version 29 (IBM Corp, Armonk, NY, USA). The data analysis involved expressing continuous variables as medians and interquartile range (IQR) and categorical variables as frequencies and percentages. The 95% confidence interval (CI) for the percentage of SIV was determined using a binomial distribution. To determine the optimal age and years of employment cut-off points based on vaccination status, ROC analysis was conducted. ROC curves were used in order to convert the aforementioned continuous variables into categorical variables, facilitating the data analysis. Categorical data were analyzed using either the chi-square test or Fisher’s exact test. Normality of continuous variables was assessed using the Kolmogorov–Smirnov normality test. For comparisons, continuous variables were analyzed using either Student’s *t*-test or the Mann–Whitney U test, depending on appropriateness. The percentage of vaccinated individuals and the proportional ratio (PR) with 95% CIs were calculated for univariate analysis. Bivariate logistic regression analysis with a 95% CI was used to determine the direction of association. The selection of variables for the bivariate logistic regression model was based on factors previously reported in the literature and found to be significant in univariate analysis. All tests were two-sided, and a *p*-value of less than 0.005 was considered statistically significant.

Certain survey questions (14, 15 and 19) were scored on a five-point scale as follows: “completely disagree”, “disagree”, “neither agree nor disagree”, “agree” and “completely agree”. Replies of “completely disagree”, “disagree” or “neither agree nor disagree” were interpreted as disagreement, while replies of “completely agree” or “agree” were interpreted as agreement. Each of the survey questions 14, 15 and 19 (see Appendix A) had three sub-items. Correct answers to all three sub-questions were considered correct, whereas incorrect answers to at least one of the three sub-questions were considered incorrect. Concerning question 19 sub-question b, the right answer was completely disagree/disagree, because the COVID-19 vaccination schedule consisted of two doses of vaccinations during the study period [34].

Regarding sources of information, through univariate analysis, two groups were created. Newspapers, infectious disease committees at the health care facility, websites for the Hellenic NPHO and Ministry of Health and medical journal articles made up the first group of information sources. Television, social media platforms and general interest websites made up the second category. For each source of information, a frequency score up to 4 was counted, depending on the respondent’s answer (1 = always, 2 = often, 3 = rarely, 4 = never). To classify respondents into each group, analysis was based on the relevant frequency score.

### 2.4. Internal Consistency Reliability

By assessing Cronbach’s alpha, the questionnaire’s internal consistency was established. The reliability coefficient was calculated as 0.868, which indicates good internal consistency [35].

## 3. Results

### 3.1. Demographics

A total of 6500 health care workers were approached and 2592 completed the questionnaire and were included in the survey (response rate 39.9%). As shown in Table 1, the mean age of study respondents was 43.3 years, and the median 44.0 years. The majority of respondents were female (70.9%) and married (63.3%). Most respondents had attained a higher-level degree (higher educational institution/university: 30.9%) or Master’s/Doctorate (24.3%). The percentages of physicians and nurses were 38.1% and 39.0%, respectively.

### 3.2. Univariate and Multivariate Analysis

The total SIV coverage of respondents was 69.4% (95% CI: 67.6–71.2%). The results from univariate analysis are presented in Table 1 and Table 2. Factors negatively associated with SIV were age ≤53 years, female sex, lower educational degree, nursing occupation, midwife occupation, medical laboratory worker, administrative officer, working in the SHC sector and working in certain health districts. Years of employment <21.5 was also negatively associated with SIV, as was relying on informal sources of information regarding the COVID-19 pandemic.

Further analysis regarding sources of information showed that using television, social media channels and general interest websites as sources of information about the COVID-19 pandemic were negatively associated with SIV, as well as with knowledge/attitudes towards vaccination and COVID-19 vaccines (Table 3).

Moreover, all questions regarding knowledge and attitudes towards routine vaccination and COVID-19 vaccines were positively associated with SIV in univariate analysis. (PR: 1.31, 95% CI: 1.25–1.38, PR: 1.42, 95% CI: 1.35–1.49, PR: 1.40, 95% CI: 1.33–1.48, respectively) (Appendix A, Appendix A). Parental adherence to the national vaccination schedule for children was positively associated with SIV (PR: 2.20, 95% CI: 1.32–3.67). A positive association with SIV was also identified concerning vulnerability due to medical history of respondents or their family members (PR: 1.16, 95% CI: 1.09–1.23 and PR: 1.06, 95% CI: 1.01–1.12, respectively). Finally, COVID-19 vaccine acceptance was positively associated with SIV (Appendix A, Appendix A).

As shown in Table 4 (multivariate analysis), factors found to influence SIV included health care profession, sector of employment (primary or secondary), health district of employment, being or living with people belonging to high-risk groups due to their medical history, attitudes towards routine vaccination, knowledge of COVID-19 vaccines and COVID-19 vaccination status.

The primary reasons for SIV refusal were low risk perception (58.7%), reluctant behavior (19.3%) and fears regarding vaccine safety (8.3%). Common reasons identified for COVID-19 vaccine refusal included fears regarding vaccine safety (38.2%), inadequate information (33.8%) and low risk perception (8.5%) (Appendix A).

## 4. Discussion

According to our study results, SIV coverage for professionals in PHC and SHC facilities during the period 2020–2021 was 69.4%, based on self-reported seasonal influenza vaccine uptake. This could be considered relatively high; however, it is still far from the target set by formal recommendations for HCWs [19,25]. The Greek National Registry for seasonal influenza vaccination rates of HCWs indicated rates of 44.6% and 67.6% in the secondary and primary health care sector, respectively [24]. These percentages were the highest rates of the last five years [24]. Several countries have been concerned about SIV following the COVID-19 vaccination period, since a negative impact on routine vaccination activities was observed due to the pandemic. Therefore, authorities’ efforts were focused on increasing SIV coverage for both HCWs and the general population in order to avoid seasonal influenza and COVID-19 coinfections [36]. A great deal of research showed a vast increase in SIV intention during the COVID-19 period, especially after initiating the COVID-19 vaccine roll-out [21,37,38,39]. Further research could examine the SIV increase as a result of the population’s increased adherence to public health recommendations due to the pandemic threat compared to previous non-pandemic years.

Although a statistically significant association was not established between SIV and education level, our study revealed indications of a dose–response effect, as the potential for SIV increased gradually and proportionally to the level of educational degree. Moreover, indirect indications of these proportional associations could be identified through occupational comparison of physicians versus nurses (six years of education versus four years, respectively). Similar findings were observed in another study conducted by our research team concerning COVID-19 vaccination in Greece [29]. Generally, education level plays an important role in SIV, as many studies have demonstrated that participants holding an MSc and/or PhD degree were more likely to accept SIV [40,41].

Occupation has proved to be an important factor associated with SIV acceptance in our study, as nurses and administrative officers reported lower SIV coverage than physicians. This confirms the results of other surveys demonstrating greater hesitancy of nurses toward SIV vaccination compared with physicians [17,42,43]. Moreover, several studies noted that physicians are more vaccine-acceptant, whether related to SIV or any other vaccine [28,29,39,44,45,46,47,48]. Similarly, a Greek survey conducted during the H1N1 pandemic examining HCWs’ uptake of the H1N1 vaccine revealed that physicians were more acceptant compared to other HCWs [49]. Conversely, nurses are one of the most vaccine-resistant occupational groups. Similar findings were observed in studies related to COVID-19 vaccination [28,29,50,51]. This behavior demonstrates a lack of risk perception ability for this specific occupational group, which could suggest hazards for both themselves and their patients. A recent Greek survey depicted the low risk perception of nurses regarding seasonal influenza [52]. Assessing both graduate and postgraduate educational curricula of nurses’ institutions could be considered, with a focus on public health and occupational ethical obligations, to achieve successful behavior change towards SIV in this specific target group.

In our study, personnel working in the PHC sector presented a higher vaccination rate than those working in hospitals (SHC). A study conducted in Poland concluded that health professionals in PHC facilities were more acceptant of SIV compared to HCWs in hospitals [47]. Seemingly, PHC professionals’ qualifications and training—which are related to vaccination policies and risk perception of vaccine-preventable diseases—contribute most to higher SIV acceptance. However, an interesting finding is that respondents of certain health districts (Attica and southern Greece) reported higher SIV rates. A possible explanation is that apart from professional qualifications and training, the dissemination of information and SIV encouragement initiatives could play a decisive role. Unfortunately, our study did not examine the implementation of SIV-related policies in each health district, or the form of information dissemination, in order to confirm this theory.

According to our study findings, respondents experiencing vulnerability due to their own or a family members’ medical issues reported higher SIV coverage. Among a number of important reasons described for SIV in several studies, HCWs stated the desire to protect themselves and their families from the disease [15,45,46,53]. Moreover, research data support the suggestion that HCWs suffering from chronic medical conditions or living with people facing medical issues were highly motivated regarding SIV [37]. Importance of disease prevention measures should be widely communicated by physicians in this specific group, and this could be a possible explanation for our findings. However, two studies conducted by our research team involving HCWs and COVID-19 vaccination observed no differences concerning vaccination status between vulnerable and non-vulnerable participants [28,29]. The novelty of COVID-19 vaccines may affect participants’ behavior, as the primary reason for refusal was fear related to vaccine safety.

Survey respondents with positive attitudes towards vaccination as well as a satisfactory knowledge of vaccination and COVID-19 vaccines had greater odds of being vaccinated for seasonal influenza. Adequate knowledge and positive attitudes towards vaccination are proven facilitators of vaccine acceptance [15,20,48,54]. It is evident that vaccination policies should include training initiatives, providing knowledge and behavior change tools in order to accomplish their purpose of improving vaccination coverage.

Our results suggested that respondents vaccinated for COVID-19 were more likely to be vaccinated for seasonal influenza. Multiple surveys agreed with our findings, and in general, SIV and COVID-19 vaccination appear to have a two-way association in the literature; COVID-19-vaccinated HCWs were more SIV-acceptant and vice versa [21,22,28,29,39]. As mentioned, it is likely that positive vaccination behavior could apply to the acceptance of any approved vaccine that is formally recommended.

According to our survey, respondents who obtained their information from formal sources such as authority websites and the medical literature were more prone to SIV. Data from the international literature support our results [55,56]. As hesitant individuals tend to draw information from informal sources, initiatives to combat vaccination hesitancy should focus on wide dissemination of adequate and documented information through informal communication channels. However, further research in this field is required.

The primary reason for SIV refusal in our survey was an underestimation of disease risk, a finding which was presented in previous related studies in Greece [57]. However, during the H1N1 pandemic, a Greek study concerning vaccination with the then “novel” H1N1 seasonal influenza vaccine reported that vaccine-hesitant HCWs cited safety issues, even though vaccination referred to a well-known disease [49]. Similarly, HCWs reported concerns about COVID-19 vaccine safety as the main reason for vaccine refusal. Many international surveys have depicted aspects of similar results regarding SIV and COVID-19 vaccination [15,46,47,53,58,59,60,61]. Studies have also demonstrated that different occupational groups may require different approaches to overcome vaccine hesitancy [44]. Ultimately, it is likely that each vaccine and vaccine-related disease may require specific and individualized policies to increase vaccine acceptance.

Our study has several limitations. The convenient sample and relatively low response rate (39.87%) could be considered as a limitation of the study related to generalization of the results. Moreover, it should be noted that the questionnaire was not externally validated for the appropriateness of the KAP questions. However, internal validity was examined with Cronbach’s alpha test. Participation bias could have occurred, as unvaccinated individuals may have been less willing to participate in the survey, even though anonymity was ensured. However, the observed vaccine acceptance (69.4%) was comparable with the HCW vaccination coverage reported by the National Registry for SIV [24]. Due to peculiarities of the specific time period—a pandemic, initiation of COVID-19 vaccines and respondents’ work overload—the questionnaire length was limited. Moreover, relevant SIV issues may not be addressed, for example, vaccination history, trust in authorities and pharmaceuticals, religious aspects, etc. [32]. Despite the aforementioned limitations, our study value comes from the nationwide nature of the sample, during a period with great impact on the vaccination schedule for seasonal influenza globally.

## 5. Conclusions

The SIV rate among HCWs in Greece can be considered relatively high. While it remains lower than the WHO recommendation of 75%, the observed rate was the highest seen over the last five years. However, certain professional categories such as nurses remain skeptical. Hesitant HCWs tend to have low risk perception related to seasonal influenza disease. Efforts should be made to address SIV hesitancy appropriately and efficiently, leading to a more targeted approach.

## Figures and Tables

**Table 1 ijerph-20-06247-t001:** Demographic characteristics associated with SIV acceptance, expressed with proportional ratio (PR) in univariate analysis (N = 2592).

Variables		Total N (%) or Median (IQR)	Vacinated N (%) or Median (IQR)	Proportional Ratio (95% CI)	Significance
Age	≤53	2138 (82.6)	1454 (68.0)	0.89 (0.84–0.95)	<0.001 (C)
	>53	451 (17.4)	343 (76.1)	Ref.	
Gender	Female	1838 (70.9)	1252 (68.1)	0.94 (0.89–0.99)	0.026 (C)
	Male	754 (29.1)	547(72.5)	Ref.	
Marital status	Divorced	95 (3.7)	64 (67.4)	0.96 (0.83–1.11)	0.567 (C)
	Widowed	7 (0.3)	3 (42.9)	0.61 (0.26–1.44)	0.207 (F)
	Unmarried	794 (30.6)	534 (67.6)	0.96 (0.91–1.02)	0.208 (C)
	N/A	55 (2.1)	44 (80.0)	1.14 (1.00–1.31)	0.115 (C)
	Married	1641 (63.3)	1151 (70.1)	Ref.	
Educational level	High school	177 (6.8)	86 (48,6)	0.61 (0.52–0.71)	<0.001 (C)
	Institute of vocational training (IEK)	216 (8.3)	119 (55.1)	0.70 (0.61–0.78)	<0.001 (C)
	Technological educational institute (TEI)	797 (29.6)	459 (59.8)	0.75 (0.70–0.80)	<0.001 (C)
	Master’s/Doctorate	631 (24.3)	495 (78.9)	0.98 (0.93–1.04)	0.471 (C)
	Higher education institute/university (BSc, AEI)	800 (30.9)	640 (80.0)	Ref.	
Health care profession	Health consultant	57 (2.2)	49 (86.0)	1.03 (0.93–1.15)	0.610 (C)
	Administrative	178 (6.9)	108 (60.7)	0.73 (0.64–0.82)	<0.001 (C)
	Medical laboratory worker	85 (3.3)	49 (57.6)	0.69 (0.58–0.83)	<0.001 (C)
	Midwife	55 (2.1)	38 (69.1)	0.83 (0.69–0.99)	0.006 (C)
	Health promotion specialist	50 (1.9)	41 (82.0)	0.98 (0.86–1.12)	0.798 (C)
	Other health professionals	170 (6.6)	102 (60.0)	0.72 (0.63–0.82)	<0.001 (C)
	Nursing staff	1010 (39.0)	589 (58.3)	0.70 (0.66–0.74)	<0.001
	Physician	987 (38.1)	823 (83.4)	Ref.	
Sector of employment	RHC	983 (37.9)	759 (77.2)	0.84 (0.71–1.00)	0.317 (F)
	UHU	153 (5.9)	139 (90.9)	0.99 (0.83–1.18)	0.999 (F)
	Public hospital	1444 (55.7)	890 (61.6)	0.67 (0.56–0.80)	0.036 (C)
	Private hospital	12 (0.5)	11 (91.7)	Ref.	
Sector of employment	Secondary health care	1456 (56.2)	901 (61.9)	0.71 (0.55–0.91)	0.002 (C)
	Primary health care	1136 (43.8)	898 (79.0)	Ref.	
Health district of employment	1st	189 (7.3)	146 (77.2)	0.91 (0.83–1.01)	0.060 (C)
	3rd	442 (17.1)	268 (60.6)	0.72 (0.65–0.79)	<0.001 (C)
	4th	512 (19.8)	329 (64.3)	0.76 (0.70–0.83)	<0.001 (C)
	5th	602 (23.2)	395 (65.6)	0.78 (0.72–0.84)	0.001 (C)
	6th	385 (14.9)	297 (77.1)	0.91 (0.84–0.99)	0.030 (C)
	7th	248 (9.6)	183 (73.8)	0.87 (0.79–0.96)	0.005 (C)
	2nd	214 (8.3)	181 (84.6)	Ref.	
Health districts of employment (groups)	3,4,5 (<70%)	1556 (60.0)	992 (63.8)	0.82 (0.78–0.86)	<0.001 (C)
	1,2,6,7 (>70%)	1036 (40.0)	807 (77.9)	Ref.	
Years of practice	<21.5	1831 (71.3)	1248 (68.2)	0.94 (0.89–0.99)	0.026 (C)
	≥21.5	738 (28.7)	536 (72.6)	Ref.	

(C): chi-square test; (F): Fisher’s exact test.

**Table 2 ijerph-20-06247-t002:** Results of association between sources of information and influenza vaccination acceptance.

Variables		Total N (%)	Vaccinated (%)	Proportional Ratio (95% CI)	Significance
Which channels do you use to keep informed about the COVID-19 pandemic and the SARS-CoV-2 vaccine, and how often?	Television, social media channels, general interest websites	1039 (40.6)	630 (60.6)	0.81 (0.76–0.85)	<0.001 (C)
	Medical articles in journals, infectious diseases committees at the health facility, website of the Hellenic National Public Health Organization (NPHO), website of the Hellenic Ministry of Health, newspapers	1519 (59.4)	1144 (75.3)	Ref.	

(C): chi-square test.

**Table 3 ijerph-20-06247-t003:** Results of association between sources of information and questions related to knowledge/perceptions about vaccination (questions 14, 15), COVID-19 vaccines (question 19) and SIV acceptance (question 17) (Appendix A—questionnaire).

Sources of Information	Proportional Ratio (PR) with 95% CI			
(Always/often vs. rarely/never)	Q14 Vaccination knowledge (0 incorrect vs. >1 incorrect)	Q15 Vaccination attitudes (0 incorrect vs. >1 incorrect)	Q19 COVID-19 vaccine knowledge (0 incorrect vs. >1 incorrect)	Q17 SIV acceptance
Television	0.65 (0.55–0.76)	0.74 (0.66–0.82)	0.76 (0.70–0.82)	0.94 (0.89–0.99)
Social media	0.80 (0.73–0.89)	0.73 (0.66–0.82)	0.83 (0.76–0.89)	0.92 (0.87–0.97)
Newspapers (print or electronic versions)	1.12 (1.02–1.24)	1.00 (0.91–1.11)	1.16 (1.07–1.25)	1.09 (1.03–1.14)
General interest publications/journals	0.90 (0.80–1.01)	0.91 (0.81–1.02)	0.96 (0.88–1.05)	1.02 (0.96–1.08)
Medical articles in journals	1.78 (1.57–2.03)	2.20 (1.91–2.54)	1.80 (1.62–2.00)	1.32 (1.24–1.41)
Infectious diseases committees at health facility	1.11 (1.00–1.22)	1.16 (1.05–1.28)	1.19 (1.10–1.29)	1.15 (1.09–1.21)
General interest websites	0.88 (0.79–0.98)	0.83 (0.74–0.92)	0.84 (0.77–0.91)	0.94 (0.89–0.99)
Hellenic National Public Health Organization (NPHO)	1.85 (1.62–2.10)	1.57 (1.38–1.77)	1.48 (1.34–1.63)	1.28 (1.20–1.36)
Website of the Hellenic Ministry of Health	1.69 (1.53–1.88)	1.39 (1.25–1.54)	1.31 (1.21–1.42)	1.25 (1.19–1.32)

**Table 4 ijerph-20-06247-t004:** Factors associated with SIV acceptance, expressed with adjusted odds ratio (aOR), in multivariable analysis.

Variables		aOR 95% CI	Significance
Age (≥53/<53)		0.79 (0.57–1.10)	0.159
Gender (male/female)		1.03 (0.99–1.30)	0.772
Educational level	High school	0.64 (0.40–1.01)	0.053
	Institute of vocational training (IEK)	0.81 (0.52–1.26)	0.355
	Technological educational institute (TEI)	0.82 (0.57–1.17)	0.276
	Master’s/Doctorate	1.09 (0.80–1.49)	0.576
	Higher education institute/university (BSc, AEI)	Ref.	
Health care profession	Nursing staff	0.66 (0.46–0.93)	0.018
	Administrative	0.59 (0.39–0.91)	0.017
	Health consultant	1.16 (0.50–2.69)	0.736
	Medical laboratory worker	0.92 (0.52–1.63)	0.770
	Midwife	0.72 (0.36–1.46)	0.360
	Health promotion specialist	1.04 (0.45–2.39)	0.934
	Other health professionals	0.56 (0.35–0.89)	0.015
	Physician	Ref.	
Sector of employment	Secondary health care	0.59 (0.46–0.75)	<0.001
	Primary health care	Ref.	
Health district of employment (groups)	3,4,5/1,2,6,7	0.66 (0.53–0.81)	<0.001
Years of practice	≥21.5 <21.5	0.78 (0.60–1.02)	0.074
Do you belong to a vulnerable/high-risk group due to your medical history? (yes/no)		1.84 (1.38–2.46)	<0.001
Do you live with older individuals or individuals belonging to a vulnerable/high-risk group due to their medical history? (yes/no)		1.46 (1.16–1.82)	0.001
Q14 (0 incorrect/>1 incorrect) Vaccination knowledge question		1.13 (0.87–1.46)	0.353
Q15 (0 incorrect/>1 incorrect) Vaccination attitudes question		1.86 (1.46–2.36)	<0.001
Source of information	Television, social media channels, general interest websites	0.83 (0.68–1.02)	0.078
	Medical articles in journals, infectious diseases committees at the health facility, website of the Hellenic National Public Health Organization (NPHO), website of the Hellenic Ministry of Health, newspapers	Ref.	
Q19 (0 incorrect/>1 incorrect) COVID-19 vaccine knowledge		1.53 (1.23–1.90)	<0.001
Q20 COVID-19 vaccination or vaccination intention (yes/no)		3.45 (2.72–4.39)	<0.001

aOR: adjusted odds ratio, Q: question.

## Data Availability

Not applicable.

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
