# Peer review of "Nationwide Survey on Seasonal Influenza Vaccination among Health Care Workers during the COVID-19 Pandemic in Greece: Determinants, Barriers and Peculiarities"

_ijerph, 2023, doi:10.3390/ijerph20136247_

Round 1
Reviewer 1 Report
this is a well-written article addressing the factors associated with influenza vaccine hesitancy among health care workers which will help with interventions to address the hesitancy and increased vaccine uptake among healthcare workers specially given their important role in addressing vaccine uptake among the general population
consider changing the s2 figure1 title within the graph to an English version.
Reviewer 2 Report
Ioanna Avakian et al carried out a cross-sectional survey to evaluate the association of a series of variables to the attitude to SIV during the Covid-19 pandemic within an HCWs population. The study potentially is significant in terms of information for public health managers and policymakers.
In general, the study is well-designed. However, a few aspects need to be pointed out.
First, the proposed questionnaire, though evaluated for internal consistency, lacks external validation especially for the correct appropriateness of KAP questions, unless they have been already validated by some others, and to the best of my knowledge, they are not. If what is said is correct, the authors should explicitly state this limitation in the paper.
A second point is about the response rate. The authors state that the response rate can be considered acceptable (line 344), given the fact it is a hybrid format, supporting this statement with references 63 and 64. Actually, those references are not useful to support a kind of cut-off of acceptability in the response rate. On the contrary, the study is based on a rate that does not allow any generalization of the conclusion of the study, unless the absence of any difference in demographic, psychosocial, and other factors among responders and non-responders could be demonstrated. That's why I strongly suggest addressing this aspect as a limitation of the study, more precisely than what it is now.
Author Response
Please see the attacment

Reviewer 3 Report
Manuscript “Nationwide survey on seasonal Influenza vaccination among health care workers during the COVID-19 pandemic in Greece: determinants, barriers and peculiarities”
Thank you for the opportunity to review this manuscript. The article addresses the interesting issue of influenza vaccination among health care workers during the pandemic situation and their possible vaccination scepticism in relation to the sources of information and occupational groups.
The text is well structured, nevertheless I have some comments:
Abstract:
For better overview and easier reading, fewer numbers would be helpful. The listing of risk factors etc. would be sufficient. If necessary, the results can be read in the result section.
Which occupational group is especially sceptical?
2. Can the questionnaire be found in the supplement? The description with question xy is inappropriate and not comprehensible if the questionnaire is not available.
3. The results should be revised. I did not find this paragraph easy to read and that is a pity because the results are important. Some of the tables do not have a heading and are perhaps also difficult to understand (maybe due to the formatting). The text sometimes lacks reference to the results in the table and the numbers are repeated.
It should be considered whether all results are really relevant and need to be reported?
Would key points instead of the entire questions in the table perhaps be clearer? And the question numbers are not comprehensible.
